# Networked Information Aggregation for Binary Classification

**MohammadHossein Bateni** [1]   **Zahra Hadizadeh** [2]   **MohammadTaghi Hajiaghayi** [3]   **Mahdi JafariRaviz** [3]
**Shayan Taherijam** [2]

## Abstract

We study networked binary classification on a directed acyclic graph (DAG), where each agent observes only a subset of the feature coordinates. Agents act sequentially along the DAG: each receives logits from its parents, augments its local features with these logits, fits a logistic predictor by minimizing binary cross-entropy (BCE), and forwards its logit to its outgoing neighbors. We ask whether this sequential distributed protocol achieves *information aggregation*: can a downstream agent attain small excess loss compared to the best logistic predictor with access to all features? This question was studied for linear regression under squared loss by Kearns, Roth, and Ryu (Kearns et al., 2026). Extending their guarantees to classification is nontrivial because their squared-loss tools do not directly transfer to BCE with a logistic link. We analyze the resulting sequential logit-passing protocol and prove an excess-loss upper bound of $O(M/\sqrt{D})$ on depth-$D$ paths satisfying the $M$-coverage condition, namely that every contiguous block of $M$ agents collectively observes all features. We also prove complementary hard instances with excess loss at least $\Omega(k/D)$, where $k$ is the feature dimension. Together, these results show that network depth is a fundamental bottleneck for information aggregation in networked logistic regression.

## 1. Introduction

The study of *social learning in networks* addresses a fundamental question in distributed learning: How do agents with partial and heterogeneous information aggregate their observations to form an accurate global belief? This line

of inquiry has a rich literature in economics and network science, beginning with the seminal work of DeGroot (DeGroot, 1974). The DeGroot model conceptualizes learning as an iterative process of weighted averaging, where agents update their scalar beliefs based on the beliefs of their neighbors. This heuristic approach was subsequently refined by Bayesian models of observational learning and information cascades, where rational agents infer private signals from the actions of their predecessors to reach a consensus or truth. For example, Banerjee (Banerjee, 1992) studies herding behavior in sequential decision-making; Bikhchandani et al. (Bikhchandani et al., 1992) formalize informational cascades as a mechanism for fads and fashions; Gale and Kariv (Gale & Kariv, 2003) analyze Bayesian learning dynamics over social networks; and Golub and Jackson (Golub & Jackson, 2010) study conditions under which naive averaging aggregates information.

While classical social learning focuses on the aggregation of scalar estimates for a single state variable, modern applications increasingly demand *networked machine learning*, where agents collaboratively learn high-dimensional hypothesis functions. In this setting, aggregation entails reconstructing a complex predictive relationship—such as a classifier mapping a high-dimensional feature vector to a label—from information dispersed across a network. Recently, Kearns et al. (Kearns et al., 2026) introduced a formal framework for this problem, embedding learning agents in a directed acyclic graph (DAG). In their protocol, agents observe a local subset of features and the predictions of their parents, training a model to minimize a local loss function. For linear regression under squared loss, they showed that such a process can allow agents to achieve excess loss competitive with a global learner having access to all features, with network depth acting as the critical resource for aggregation.

The DAG formulation is useful beyond inherently acyclic communication networks. Many multi-round protocols can be represented by "unrolling" time: one creates a copy of each active agent in each round and connects these copies according to the communication pattern between consecutive rounds. Under this interpretation, the depth of the DAG corresponds to the number of sequential communication and training steps available to the protocol. Thus, path-based

[1]Google Research, New York City, NY, USA [2]University of California, Irvine, CA, USA [3]University of Maryland, College Park, MD, USA. Correspondence to: Shayan Taherijam <staherij@uci.edu>.

constructions can also be viewed as simplified models of repeated communication over an underlying network.

## 1.1. Our Contributions

In this work, we focus on the classification variant of the model of (Kearns et al., 2026). We analyze a sequential learning protocol where agents optimize logistic regression models using local features and incoming logits from their parents in a DAG. Our main contributions are as follows.

**Upper Bounds.** We analyze information aggregation in a network of logistic regression agents. We show that if a path of length $D$ satisfies the $M$-coverage condition—namely, every contiguous block of $M$ agents collectively observes all features—then the excess loss of the final agent scales as $O(M/\sqrt{D})$ (Theorem 3.8).

At a high level, our proof extends the analysis of (Kearns et al., 2026) from squared loss to logistic loss. The squared-loss argument relies on an exact variance decomposition, which does not carry over to Binary Cross-Entropy (BCE). Instead, we certify progress via a KL/Bregman-type characterization of loss differences (Lemma 3.3) together with a Pinsker-style link from KL progress to prediction error (Lemma 3.4). A key technical input is an orthogonality condition for BCE residuals (Lemma 3.1). Finally, a stability argument identifies a segment of the path where improvement saturates; this forces small residuals and yields a bound on the deviation from the global optimum.

**Lower Bounds and Hard Instances.** We complement the upper bound by exhibiting a hard instance for the sequential logit-passing protocol. The construction is designed so that most individual features are uninformative about the label in isolation, while the full set of features reveals the relevant latent signal through a simple linear relation. Thus, missing some features leaves a nontrivial amount of noise that cannot be removed by the restricted sequential protocol. We prove that the excess loss is lower-bounded by $\Omega(k/D)$, where $k$ is the dimension of the feature space.

Together, the upper and lower bounds show that network depth is a genuine bottleneck for information aggregation in networked logistic regression. The upper bound shows that sufficient depth guarantees aggregation under the $M$-coverage condition, while the lower bound shows that depth is also necessary for some instances. There remains a gap between the upper bound $O(M/\sqrt{D})$ and the lower bound $\Omega(k/D)$, which we leave as an interesting direction for future work.

## 1.2. From Regression to Classification

Extending theoretical guarantees of networked aggregation from linear regression to binary classification is nontrivial. The analysis in (Kearns et al., 2026) relies fundamentally on the geometry of squared loss—in particular, orthogonality of residuals and a Pythagorean variance decomposition—which translate loss reduction directly into parameter-space convergence. In contrast, binary classification via logistic regression and Binary Cross-Entropy (BCE) does not admit a comparable bias–variance decomposition. The nonlinearity of the sigmoid link also creates an important modeling issue: linear aggregation in probability space is not equivalent to linear aggregation in feature space. For this reason, our protocol passes logits rather than probabilities; logits remain linear functions of the available features and previously received logits, whereas probabilities after the sigmoid transformation do not.

Despite these challenges, classification is a central task in distributed and networked learning, including settings such as medical diagnosis (Vepakomma et al., 2018) and vertically partitioned prediction problems. Establishing guarantees for information aggregation under BCE is therefore a natural theoretical goal.

More broadly, this regression-to-classification gap is not unique to our networked setting: across several areas, techniques and guarantees that are clean for *squared-loss regression* require different tools once the target is *probabilistic classification*. We list three representative pairs. First, in *sketching/subspace-embedding* methods, least-squares regression admits sharp oblivious sketching guarantees (Clarkson & Woodruff, 2013), whereas logistic objectives require non-$L_2$ progress measures and different proof techniques (Munteanu et al., 2021). Second, in *second-order / curvature-aware* acceleration via sketching, iterative Hessian sketch methods are cleanly analyzed for constrained least squares (Pilanci & Wainwright, 2016), while Newton-sketch extensions for regularized ERM objectives, including logistic regression, require controlling curvature that depends on current predictions (Pilanci & Wainwright, 2017). Third, in *distribution-free predictive inference*, split conformal prediction yields regression intervals (Lei et al., 2018), whereas classification requires set-valued prediction regions and different uncertainty objects (Angelopoulos & Bates, 2021). Taken together, these examples support the message most relevant to our paper: moving from regression to classification typically replaces Euclidean residual decompositions with KL/Bregman-flavored notions of progress, which becomes unavoidable when the information flowing through the network is itself a low-bandwidth probabilistic prediction.

## 1.3. Related Work

The most directly related body of work comes from *Vertical Federated Learning (VFL)*, where different parties hold disjoint feature columns for the same aligned examples and collaborate to train a joint predictor. This feature-partitioned

setting is close to ours, but the standard VFL setting is inherently *interactive*: the survey of (Yang et al., 2019) formalizes the feature-partitioned regime and highlights that most practical protocols rely on repeated message exchanges (e.g., gradients, activations, or protected sufficient statistics) to optimize a shared objective. In contrast, our model studies a DAG-based logit-passing protocol, where agents receive predictions/logits from their parents and pass only their own learned logit downstream.

This view is reflected in deployed platforms such as FATE (Liu et al., 2021), which operationalize multi-round VFL pipelines and make explicit the practical tension between accuracy, privacy protection, and communication cost. A particularly relevant algorithmic family is *vertical* gradient-boosted tree training: SecureBoost (Cheng et al., 2021) and later high-performance variants such as SecureBoost+ (Fan et al., 2024) show that strong predictors can be trained over vertically split features, but only by repeatedly coordinating split decisions through exchanging split-related information. From the perspective of our setting, these works provide concrete evidence that *feature partitioning* alone already induces a strong communication bottleneck; our work asks what can be guaranteed when the interaction budget is pushed much closer to its limit.

A less direct but relevant comparison is *split learning* (Vepakomma et al., 2018), which was proposed for distributed settings such as health care where parties train models without sharing raw data. In split learning, parties communicate intermediate neural-network activations and gradients rather than raw features. This differs from our logit-passing model, but it reflects a similar concern: the choice of what representation is transmitted can strongly affect both communication and prediction quality.

Finally, recent surveys (Wu et al., 2025; Khan et al., 2025) synthesize the VFL literature and stress that communication (both message size and number of rounds) remains a dominant practical limitation in VFL, alongside privacy leakage and the statistical dependence patterns induced by feature/label partitioning. This framing closely matches the communication-centric viewpoint taken in our analysis, although our technical model is different from standard VFL: we do not design an interactive federated optimization protocol, but instead analyze what can be learned from sequential logit passing over a network.

## 2. Preliminaries

In this section, we formally define the notation, the logistic regression framework, and the distributed learning setup.

### 2.1. Binary Classification and Logistic Regression

We consider a binary classification problem. Let $\mathcal{D}$ be a distribution over feature-label pairs $(x, y)$, where $x \in \mathbb{R}^d$ is a vector of features and $y \in \{0, 1\}$ is the binary target label.

We model the conditional probability $P(y = 1|x)$ using the logistic function $\sigma(z) = \frac{1}{1+e^{-z}}$. For a probability $p \in (0, 1)$, its logit is $\log(p/(1 - p))$; equivalently, if $p = \sigma(z)$, then $z$ is the logit. A hypothesis is parameterized by a vector $\theta \in \mathbb{R}^d$, yielding the predictor:

$$p^{(\theta)}(x) = \sigma(\theta^T x). \tag{1}$$

The quality of a predictor is measured by the expected Binary Cross Entropy (BCE) loss:

$$L(\theta) = -\mathbb{E}_{(x,y)\sim\mathcal{D}}[y \log p^{(\theta)}(x) + (1-y) \log(1-p^{(\theta)}(x))]. \tag{2}$$

The global Maximum Likelihood Estimator (MLE), denoted by $p^*$, corresponds to the parameters $\theta^*$ that minimize this loss over the full feature space. Throughout the paper, we assume that the relevant logistic-loss minimizers exist; in particular, the full-information optimum $p^*$ is attained.

### 2.2. Distributed Learning Setup

We consider a distributed learning setting with a set of $N$ agents, $\mathcal{A} = \{A_1, \ldots, A_N\}$. The agents are organized in a Directed Acyclic Graph (DAG), $G = (\mathcal{A}, E)$, where an edge $(A_j, A_i) \in E$ indicates that agent $A_i$ receives information from agent $A_j$. We sometimes also write $A_j \to A_i$ to denote this relationship. We denote the set of parents of agent $A_i$ as $\text{Pa}(A_i) = \{A_j \mid (A_j, A_i) \in E\}$. The agents learn in an order consistent with a topological sort of the DAG, with ties in the topological ordering broken arbitrarily.

Let $[d] = \{1, 2, \ldots, d\}$ be the set of indices for $d$ total features. Each agent $A_i \in \mathcal{A}$ is associated with a specific subset of these features, $S_i \subseteq [d]$. For any agent $A_i$, its local view of the features is $x_{S_i}$, which is the sub-vector of $x$ corresponding to the features indexed by $S_i$. The agent also receives its parents' logits.

### 2.3. Sequential Learning Protocol

The agents learn models in a sequential manner. Unlike linear regression settings where agents minimize squared error, here each agent $A_i$ aims to train a model $f_i$ to minimize the local Binary Cross Entropy (BCE) loss.

Each agent $A_i$ observes its local features $x_{S_i}$ and the set of outputs from its parents. Agents communicate their logits rather than their final probabilities, so the transmitted messages remain linear functions of the available features and previously received logits before applying the sigmoid function. Let $z_j$ be the logit output by parent $A_j$, such that the parent's prediction is $\hat{p}_j = \sigma(z_j)$.

**Algorithm 1** Sequential logit-passing protocol

---

**Require:** DAG $G = (\mathcal{A}, E)$, feature sets $\{S_i\}_{i=1}^N$, distribution $\mathcal{D}$
**Ensure:** Final prediction of a sink agent
1: Choose any topological ordering of the agents.
2: **for** each agent $A_i$ in this order **do**
3:     Receive parent logits $\{z_j(x) : A_j \in \mathrm{Pa}(A_i)\}$.
4:     Form the augmented input

$$\left(x_{S_i}, \{z_j(x)\}_{A_j \in \mathrm{Pa}(A_i)}\right).$$

5:     Choose parameters $w_i$ and $\{v_{ij}\}_{A_j \in \mathrm{Pa}(A_i)}$ minimizing the BCE loss $L(p_i)$, where

$$p_i(x) = \sigma\left(w_i^T x_{S_i} + \sum_{j \in \mathrm{Pa}(A_i)} v_{ij} z_j(x)\right).$$

6:     Set $z_i(x) = w_i^T x_{S_i} + \sum_{j \in \mathrm{Pa}(A_i)} v_{ij} z_j(x)$ and send $z_i$ to the children of $A_i$.
7: **end for**
8: **return** the prediction $p_i = \sigma(z_i)$ of a sink agent.

---

The model $p_i$ for agent $A_i$ is a logistic function of its local features and the parents' logits:

$$p_i(x) = \sigma(z_i(x)) \text{ where } z_i(x) = w_i^T x_{S_i} + \sum_{j \in \mathrm{Pa}(A_i)} v_{ij} z_j(x).$$
$$(3)$$

Here, $w_i$ (weights for local features) and $v_{ij}$ (weights for incoming logits) are learnable parameters. Agent $A_i$ chooses these parameters to minimize the BCE loss $L(p_i)$; the full protocol is summarized in Algorithm 1.

The final output of the system is the prediction of the sink agent (or the last agent in the topological sort).

We write $p_i$ for the prediction function $x \mapsto p_i(x)$ when no confusion arises. We use the notation $L(p)$, $L(\theta)$, and $L(z)$ interchangeably when they represent the same predictor. Here, $\theta$ denotes the weight vector. We define $z(x) = \theta^T x$ and $p(x) = \sigma(z(x))$.

## 3. Upper Bounds

This section analyzes information aggregation within a network of logistic regression agents. We demonstrate that sequentially minimizing Binary Cross Entropy (BCE) allows the network to approximate the global predictor derived from all features, assuming sufficient network depth and feature coverage.

We first establish that the residuals of the BCE loss optimizer are orthogonal to the input. A similar result was previously derived by (Kearns et al., 2026) for the linear regression with MSE loss.

**Lemma 3.1** (Orthogonality of Residuals). *Assume that the BCE minimizer over the feature space $\mathcal{X}$ exists and is attained at a finite parameter vector $\theta^*$. Let $p^*$ be the corresponding optimal logistic predictor. The residual error $(p^*(x) - y)$ is orthogonal to the feature vector $x$ in expectation:*

$$\mathbb{E}\left[x(p^*(x) - y)\right] = 0.$$

*Proof.* Given $p^{(\theta)}(x) = \sigma(\theta^T x)$, the gradient of the logistic output is $\nabla_\theta p^{(\theta)}(x) = p^{(\theta)}(x)(1 - p^{(\theta)}(x))x$. Applying the chain rule to $L(\theta)$ yields:

$$\nabla_\theta L(\theta) = -\mathbb{E}\left[(y(1 - p^{(\theta)}(x)) - (1 - y)p^{(\theta)}(x))x\right]$$
$$= \mathbb{E}\left[(p^{(\theta)}(x) - y)x\right].$$

By the assumed existence of a finite minimizer, the optimal parameters $\theta^*$ satisfy the first-order condition $\nabla_\theta L(\theta^*) = 0$. Thus:

$$\mathbb{E}\left[x(p^*(x) - y)\right] = 0. \qquad \square$$

This orthogonality allows us to decompose the error of any suboptimal model. We express the error of a suboptimal model in terms of the optimal predictor using the expected Kullback-Leibler divergence of the Bernoulli distribution, defined as follows.

**Definition 3.2.** Let $p$ and $q$ be two predictors on feature space $\mathcal{X}$. Then $D(p\|q)$ is defined as:

$$D(p\|q) = \mathbb{E}\left[D_{\mathrm{KL}}\left(\mathrm{Bernoulli}(p(x))\|\mathrm{Bernoulli}(q(x))\right)\right],$$

where $D_{\mathrm{KL}}(p'\|q')$ is the Kullback-Leibler divergence between distributions $p'$ and $q'$, given by:

$$D_{\mathrm{KL}}(p'\|q') = \sum_x p'(x) \log \frac{p'(x)}{q'(x)}.$$

Expanding this definition, $D(p\|q)$ becomes:

$$D(p\|q) = \mathbb{E}\left[p(x) \log \frac{p(x)}{q(x)} + (1 - p(x)) \log \frac{1 - p(x)}{1 - q(x)}\right].$$

**Lemma 3.3** (Decomposing Loss). *Let $p^*$ be the optimal logistic predictor on a feature set $S$, and let $q$ be any logistic predictor on $S$. The loss decomposes as:*

$$L(q) = L(p^*) + D(p^*\|q).$$

*Proof.* Using the identity $\log \sigma(z) = z - \log(1 + e^z)$, we write the loss with $z = \theta^T x$ as:

$$L(\theta) = -\mathbb{E}\left[yz - \log(1 + e^z)\right]$$
$$= \mathbb{E}\left[\log(1 + e^{\theta^T x}) - y(\theta^T x)\right].$$

Let $\theta^*$ be the corresponding parameters of $p^*$. Expanding the difference $L(\theta) - L(\theta^*)$:

$$L(\theta) - L(\theta^*) = \mathbb{E}\Big[\log(1 + e^{\theta^T x})$$
$$- \log(1 + e^{(\theta^*)^T x})$$
$$- y((\theta - \theta^*)^T x)\Big].$$

Adding and subtracting $p^*(x)(\theta - \theta^*)^T x$ inside the expectation yields:

$$L(\theta) - L(\theta^*) = \mathbb{E}\Big[\log(1 + e^{\theta^T x}) - \log(1 + e^{(\theta^*)^T x})$$
$$- p^*(x)(\theta - \theta^*)^T x\Big]$$
$$+ \mathbb{E}\Big[(p^*(x) - y)(\theta - \theta^*)^T x\Big]. \quad (4)$$

The second term is zero due to the orthogonality condition derived in Lemma 3.1. For the first term, we expand the definition of $D(p^*\|q)$ with $z = \theta^T x$ and $z^* = (\theta^*)^T x$:

$$D(p^*\|q) = \mathbb{E}\Big[p^*(x)(z^* - z) - \log\Big(1 + e^{z^*}\Big)$$
$$+ \log\Big(1 + e^z\Big)\Big]$$
$$= \mathbb{E}\Big[\log\Big(1 + e^{\theta^T x}\Big) - \log\Big(1 + e^{(\theta^*)^T x}\Big)$$
$$- p^*(x)(\theta - \theta^*)^T x\Big].$$

This matches the first term in Equation (4), completing the proof. $\square$

To bound the parameter error using the KL divergence, we use the following inequality. This is a specific case of Pinsker's inequality (Pinsker, 1964), included here for completeness.

**Lemma 3.4.** *For the expected KL divergence $D(p\|q)$, the following inequality holds:*

$$D(p\|q) \geq 2\mathbb{E}\left[(p(x) - q(x))^2\right].$$

We refer to Appendix A for the proof. We define the pointwise loss function as:

$$l(z, y) = \log(1 + e^z) - yz. \quad (5)$$

We can thus write $L(p) = \mathbb{E}[l(z(x), y)]$.

**Lemma 3.5.** *Let $g(x) = \sigma(z_g(x))$ be any logistic predictor. Let $S$ be a feature subspace and $p(x) = \sigma(z_p(x))$ be the predictor that minimizes $L(p)$ over $S$. Then:*

$$L(p) \leq L(g) + |\mathbb{E}[(p - y)z_g]|.$$

*Proof.* Let $\phi(z) = \log(1 + e^z)$. The derivatives are $\phi'(z) = \sigma(z)$ and $\phi''(z) = \sigma(z)(1 - \sigma(z))$. Since $\sigma(z) \in (0, 1)$, we have $\phi''(z) \geq 0$, implying $\phi$ is convex. Convexity implies that for any $u, v \in \mathbb{R}$, $\phi(v) \geq \phi(u) + \phi'(u)(v - u)$. Rearranging implies the following:

$$\phi(u) \leq \phi(v) + \sigma(u)(u - v). \quad (6)$$

We define the relationship between the losses $l(u, y)$ and $l(v, y)$. Substituting $l(z, y) = \phi(z) - yz$, we aim to show the following inequality:

$$l(u, y) \leq l(v, y) + (\sigma(u) - y)(u - v). \quad (7)$$

Expanding terms confirms this holds given the convexity of $\phi$ in Equation (6):

$$\phi(u) - yu \leq \phi(v) - yv + \sigma(u)(u - v) - yu + yv$$
$$\iff \phi(u) \leq \phi(v) + \sigma(u)(u - v).$$

Now, for a point $x$, let $u = z_p(x)$ and $v = z_g(x)$. Applying Equation (7):

$$l(z_p, y) \leq l(z_g, y) + (\sigma(z_p) - y)(z_p - z_g).$$

Taking the expectation over $x$:

$$L(p) \leq L(g) + \mathbb{E}[(p - y)(z_p - z_g)]$$
$$= L(g) + \mathbb{E}[(p - y)z_p] - \mathbb{E}[(p - y)z_g].$$

From Lemma 3.1 (Orthogonality), we know that for any feature $x_l$ in the support of $p$, $\mathbb{E}[x_l(p - y)] = 0$. Since $z_p$ is a linear combination of such features, $\mathbb{E}[(p - y)z_p] = 0$. Substituting this yields:

$$L(p) \leq L(g) - \mathbb{E}[(p - y)z_g] \leq L(g) + |\mathbb{E}[(p - y)z_g]|. \quad \square$$

We consider a path of agents $A_1, \ldots, A_D$. Each agent $i$ receives the logit $z_{i-1}$ from its predecessor and trains a logistic predictor model using locally observed features $x_{S_i}$, $z_{i-1}$, and possibly some other predecessors' logits. Since one option for the agent $A_i$ is to pass the logits $z_{i-1}$ through, we have that $L(p_{i-1}) \geq L(p_i)$. Moreover, since $p_{i-1}$ is feasible for the hypothesis class of agent $A_i$, Lemma 3.3 applied with $p_i$ as the optimizer and $p_{i-1}$ as the comparator gives

$$D(p_i\|p_{i-1}) = L(p_{i-1}) - L(p_i).$$

We use the notation $\|f(x)\|_2 = \sqrt{\mathbb{E}[f(x)^2]}$ for any function $f$.

**Lemma 3.6** (Residual Bound via Path Coverage). *Let $A_1, \ldots, A_k$ be a path of agents where every feature $x_l$ is observed at least once. Let $g(x) = \sigma(z_g(x))$ where $z_g(x) = \sum_{l=1}^d \alpha_l x_l$ be any logistic predictor over the whole space. Assume the coefficients of $z_g$ satisfy $\sum_{l=1}^d |\alpha_l| \leq B_g$,*

*and the features satisfy $\mathbb{E}[x_l^2] \leq B_X^2$, for some $B_g$ and $B_X$. Let $\varepsilon \geq L(p_1) - L(p_k)$. Then:*

$$|\mathbb{E}[(p_k - y)z_g]| \leq B_g B_X \sqrt{\frac{k\varepsilon}{2}}.$$

*Proof.* Let $z_g(x) = \sum_{l=1}^{d} \alpha_l x_l$. We bound the error term:

$$|\mathbb{E}[(p_k - y)z_g]| = \left| \mathbb{E}\left[ \sum_{l=1}^{d} \alpha_l x_l (p_k - y) \right] \right|$$

$$\leq \sum_{l=1}^{d} |\alpha_l| \, |\mathbb{E}[x_l(p_k - y)]|.$$

Consider a feature $x_l$. Due to each feature being observed, this feature appears in the index set of some agent $A_j$ in the path. By orthogonality, $\mathbb{E}[x_l(p_j - y)] = 0$. We decompose the expectation using the triangle inequality:

$$|\mathbb{E}[x_l(p_k - y)]| \leq |\mathbb{E}[x_l(p_k - p_j)]| + |\mathbb{E}[x_l(p_j - y)]|$$
$$= |\mathbb{E}[x_l(p_k - p_j)]|.$$

Applying the Cauchy-Schwarz inequality:

$$|\mathbb{E}[x_l(p_k - p_j)]| \leq \sqrt{\mathbb{E}[x_l^2]}\sqrt{\mathbb{E}[(p_k - p_j)^2]}$$
$$= \|x_l\|_2 \|p_k - p_j\|_2.$$

Given $\|x_l\|_2 \leq B_X$, we bound $\|p_k - p_j\|_2$ using the loss difference $\varepsilon$. For any $s \in \{1, \ldots, k-1\}$, Lemma 3.4 and Lemma 3.3 give

$$\|p_s - p_{s+1}\|_2 \leq \sqrt{\frac{D(p_{s+1}\|p_s)}{2}} = \sqrt{\frac{L(p_s) - L(p_{s+1})}{2}}.$$

Therefore, by the triangle inequality and Cauchy–Schwarz,

$$\|p_j - p_k\|_2 \leq \sum_{s=j}^{k-1} \|p_s - p_{s+1}\|_2$$
$$\leq \sum_{s=j}^{k-1} \sqrt{\frac{D(p_{s+1}\|p_s)}{2}}$$
$$\leq \sqrt{\frac{k \sum_{s=1}^{k-1} D(p_{s+1}\|p_s)}{2}}$$
$$= \sqrt{\frac{k(L(p_1) - L(p_k))}{2}}$$
$$\leq \sqrt{\frac{k\varepsilon}{2}}.$$

Combining these bounds with the constraint on $\alpha_l$:

$$|\mathbb{E}[(p_k - y)z_g]| \leq \sum_{l=1}^{d} |\alpha_l| \cdot |\mathbb{E}[x_l(p_k - y)]| \leq B_g B_X \sqrt{\frac{k\varepsilon}{2}}.$$
$$\square$$

We now give the below definition.

**Definition 3.7** ($M$-Coverage Condition, Kearns et al. (2026)). A path satisfies the $M$-coverage condition if every contiguous subsequence of $M$ agents collectively observes all $d$ features $x_1, \ldots, x_d$.

We are finally ready to prove Theorem 3.8. Combining Lemma 3.5 and Lemma 3.6, we obtain the relationship $L(p_k) \leq L(g) + B_g B_X \sqrt{k\varepsilon/2}$ for a path of length $k$. Extending this analysis over the full path satisfying the $M$-coverage condition leads to our main convergence result.

**Theorem 3.8** (Global Convergence Rate). *Consider a DAG $G$ containing a path of length $D$ of agents $A_1, \ldots, A_D$ satisfying the $M$-coverage condition. Let $p^*$ be the global optimal logistic predictor over all $d$ features. Assume:*

1. *Bounded second moments: $\mathbb{E}[x_l^2] \leq B_X^2$ for all $l \in \{1, \ldots, d\}$.*

2. *Bounded coefficients: for the optimal logits $z^*(x) = \sum_l \alpha_l x_l$ where $\|\alpha\|_1 \leq B_{p^*}$.*

*Then the excess risk of the final agent $p_D$ is bounded by:*

$$L(p_D) - L(p^*) \leq B_{p^*} B_X \frac{M}{\sqrt{D}} = O\left(\frac{M}{\sqrt{D}}\right).$$

*Proof.* We partition the path into $K = \lfloor D/M \rfloor$ disjoint blocks of length $M$. By the Pigeonhole Principle, since the total loss reduction is bounded by the loss of the first agent $L(p_1)$, there exists at least one *stable* block $k^*$ where the reduction is at most the total reduction divided by $K$. Suppose this block $k^*$ is on indices $s, s+1, \ldots, t$.

$$\sum_{i=s+1}^{t} (L(p_{i-1}) - L(p_i)) \leq \frac{L(p_1)}{K} \leq \frac{2ML(p_1)}{D} := \varepsilon.$$

Applying Lemma 3.5 and Lemma 3.6, we get that over this path $L(p_t) \leq L(p^*) + B_{p^*} B_X \sqrt{M\varepsilon/2}$. Next, note that $L(p_1) \leq \log 2$ since using $\theta_1 = 0$ achieves a loss of $\log 2$, and because the first agent optimizes within its domain then $L(p_1) \leq \log 2 < 1$. Combined with the non-increasing losses, we get:

$$L(p_D) - L(p^*) \leq B_{p^*} B_X \frac{M}{\sqrt{D}}. \qquad \square$$

## 4. Lower Bound Analysis

In this section, we construct a theoretical lower bound for the distributed learning protocol. We demonstrate the existence of a specific data distribution and network configuration where the excess loss is at least $\Omega(k/D)$, where $D$ is the depth of the network and $k$ is the dimension of the feature space. This result confirms that the sequential nature of the protocol makes network depth a fundamental bottleneck for information aggregation.

## 4.1. Problem Construction

We define a hard instance that exploits the information bottleneck inherent in sequential logit aggregation. The intuition behind this construction is that we want the agents' features to have very limited useful information on their own. Most features are independent of the label unless they are combined with the right neighboring features. However, when all features are available together, they recover the hidden signal that determines the label. Therefore, an agent that has only aggregated part of the feature sequence still misses some information, and this missing information creates a nontrivial prediction error.

**Data Distribution.** Let $k \geq 2$ be the dimension of the feature space. Consider a sequence of independent latent variables $Z_1, Z_2, \ldots, Z_k \sim \mathcal{N}(0, 1)$. We define the observable features $x_1, \ldots, x_k \in \mathbb{R}$ as follows:

$$x_1 = Z_1 \tag{8}$$
$$x_i = Z_i - Z_{i-1}, \quad \text{for } 2 \leq i \leq k \tag{9}$$

By construction, the latent variable $Z_i$ can be recovered by the prefix sum of features: $\sum_{j=1}^{i} x_j = Z_i$. We define the binary target label $y \in \{0, 1\}$ based on the final latent variable $Z_k$ via the logistic model:

$$P(y = 1|x) = \sigma(Z_k) = \sigma\left(\sum_{j=1}^{k} x_j\right) \tag{10}$$

Thus, the optimal global logit predictor is $z^*(x) = Z_k$, which requires access to all $k$ features to cancel the intermediate noise terms.

**Network and Assignment.** Consider a path of agents $A_1, \ldots, A_D$. The agents observe features one-at-a-time in a repeating cyclic order. Agent $A_i$ observes the single feature $x_\ell$ where $\ell = ((i - 1) \pmod{k}) + 1$. The network structure is a simple path where $\text{Pa}(A_i) = \{A_{i-1}\}$. This path satisfies the $M$-coverage condition with $M = k$, since every consecutive block of $k$ agents observes all $k$ features exactly once.

We define a *pass* $p$ as the $p$-th disjoint block of $k$ agents. Specifically, the $p$-th pass consists of the agents $A_{(p-1)k+1}, \ldots, A_{pk}$.

## 4.2. Information Capacity and Variance

We analyze the capacity of the final agent in pass $p$ to reconstruct the target $Z_k$ by identifying which features can be effectively decoded from the scalar stream.

The following lemma characterizes the restricted information available to agents. This result follows the methods of Lemma 5.5 in (Kearns et al., 2026), which establishes an analogous result. We provide the proof for completeness.

**Lemma 4.1** (Recursive Information Relevance). *For any pass $p$ with $p \leq k$, define the feature subset $\mathcal{I}_p = \{x_k, x_{k-1}, \ldots, x_{k-p+1}\}$. The optimal logistic predictor for $\sigma(Z_k)$ at the end of pass $p$ depends solely on the features in $\mathcal{I}_p$.*

*Proof.* We proceed by induction on the pass index $p$.

**Base Case** ($p = 1$): The first pass ends at agent $A_k$, who observes the local feature $x_k = Z_k - Z_{k-1}$. Preceding agents $A_1, \ldots, A_{k-1}$ observe features $x_1, \ldots, x_{k-1}$, all of which are independent of the target $Z_k$ and thus the label $y$. Because each agent $A_i$ (for $i < k$) minimizes its local BCE loss using only information independent of the label, each sequentially transmits a logit of 0 to its successor. Consequently, $A_k$ receives a logit of 0 from $A_{k-1}$ and must rely exclusively on its local observation $x_k$ to predict $y$. This establishes the effective information set $\mathcal{I}_1 = \{x_k\}$.

**Inductive Step:** Assume at the end of pass $p$, the optimal predictor $z^{(p)}$ is a function only of the features in $\mathcal{I}_p = \{x_k, x_{k-1}, \ldots, x_{k-p+1}\}$. In pass $p+1$, the initial sequence of agents observe features $x_1, \ldots, x_{k-p-1}$. Because these features are independent of the current information set $\mathcal{I}_p$, they are also independent of the incoming logit $z^{(p)}$ and the label $y$. Consequently, these agents cannot improve the prediction; they sequentially forward the logit $z^{(p)}$ to one another without modification. This process continues until an agent observes $x_{k-p} = Z_{k-p} - Z_{k-p-1}$. This new feature is correlated with the latent variable $Z_{k-p}$ currently acting as noise in $z^{(p)}$, allowing the agent to partially cancel that noise and improve the estimate of $Z_k$. Subsequent agents in the pass observe only features in $\mathcal{I}_p$. Thus, the relevant information set expands by exactly one feature: $\mathcal{I}_{p+1} = \mathcal{I}_p \cup \{x_{k-p}\}$. $\square$

Given this restriction, any logit $z^{(p)}$ generated at the end of pass $p$ is a linear function of the features in $\mathcal{I}_p$. We analyze this linear predictor in the following lemma.

**Lemma 4.2.** *Let $z^{(p)}$ be a linear predictor based on $\mathcal{I}_p$ whose coefficient on $Z_k$ is $c \neq 0$. Then $z^{(p)}$ is given by*

$$z^{(p)} = c\left(Z_k + \frac{1}{\sqrt{p}}\xi\right),$$

*where $\xi \sim \mathcal{N}(0, V_p)$ is independent of $Z_k$. For a fixed coefficient $c$ on $Z_k$, the minimum possible residual variance is $c^2/p$; equivalently, in the above normalization, the minimum possible value of $V_p$ is 1.*

*Proof.* Define $z^{(p)} = \sum_{j=0}^{p-1} c_j x_{k-j}$ with coefficients $c_0, \ldots, c_{p-1}$. We rewrite this expression in terms of the variables $Z_k, \ldots, Z_{k-p}$:

$$z^{(p)} = c_0 Z_k + \sum_{j=1}^{p-1}(c_j - c_{j-1})Z_{k-j} - c_{p-1}Z_{k-p}.$$

Let $c = c_0$ and $\alpha_j = c_j - c_{j-1}$. Substituting these terms yields:

$$z^{(p)} = cZ_k + \sum_{j=1}^{p-1} \alpha_j Z_{k-j} - \left( \sum_{j=1}^{p-1} \alpha_j + c \right) Z_{k-p}.$$

Define the residual $\eta = z^{(p)} - cZ_k$. Since $Z_i$ are i.i.d. $\mathcal{N}(0,1)$, the variance of $\eta$ is $\text{Var}(\eta) = \sum_{j=1}^{p-1} \alpha_j^2 + \left( \sum_{j=1}^{p-1} \alpha_j + c \right)^2$. Define $\xi = \frac{\sqrt{p}}{c} \eta$. It follows that $\xi \sim \mathcal{N}(0, V_p)$, where $V_p = \frac{p}{c^2} \text{Var}(\eta)$. This establishes the form of the predictor.

Next, we fix $c$ and minimize $V_p$, which is equivalent to minimizing $\text{Var}(\eta)$. Let $S = \sum_{j=1}^{p-1} \alpha_j$. The variance can be written as $\text{Var}(\eta) = \sum_{j=1}^{p-1} \alpha_j^2 + (S + c)^2$. For a fixed sum $S$, the term $\sum \alpha_j^2$ is minimized when all $\alpha_j$ are equal, yielding $\text{Var}(\eta) = \frac{S^2}{p-1} + (S + c)^2$. Differentiating with respect to $S$ and setting the result to zero, we find the minimum occurs at $S = -c(p-1)/p$. Substituting this value back, the minimum variance is $c^2/p$. Consequently, $V_p = \text{Var}(\eta) \frac{p}{c^2} = 1$. $\qquad \square$

Next, we analyze the properties of the noise term $\xi$ in the linear predictor defined in Lemma 4.2 and examine its impact on the BCE loss.

**Lemma 4.3.** *Let $z_v = cZ_k + \xi_v$, where $\xi_v \sim \mathcal{N}(0, v)$ is independent of $Z_k$. For a fixed $c$, if $u > v$, then:*

$$L(z_v) < L(z_u),$$

*where $L(z)$ denotes the BCE loss.*

*Proof.* We first analyze the loss conditioned on $Z_k$. Assuming $y|Z_k \sim \text{Bernoulli}(\sigma(Z_k))$, we expand the conditional loss as:

$$L(z|Z_k) = \mathbb{E}_{y|Z_k} [-yz + \log(1 + e^z)]$$
$$= -\sigma(Z_k)z + \log(1 + e^z). \qquad (11)$$

Define $g(z) = L(z|Z_k)$. Differentiating with respect to $z$ yields $g'(z) = -\sigma(Z_k) + \sigma(z)$. The second derivative is $g''(z) = \sigma(z)(1 - \sigma(z))$. Since $\sigma(z) \in (0,1)$, $g''(z) > 0$, implying $g$ is strictly convex.

The total loss for $z_v$ is $L(z_v) = \mathbb{E}_{Z_k} [\mathbb{E}_{\xi_v} [g(cZ_k + \xi_v)]]$. To compare $z_u$ and $z_v$, let $\delta \sim \mathcal{N}(0, u - v)$ be independent of $\xi_v$. We can model the higher variance noise as $\xi_u = \xi_v + \delta$. Applying Jensen's inequality to the strictly convex function $g$:

$$\mathbb{E}_\delta [g(cZ_k + \xi_v + \delta)] > g(\mathbb{E}_\delta [cZ_k + \xi_v + \delta])$$
$$= g(cZ_k + \xi_v).$$

Taking the expectation over $Z_k$ and $\xi_v$ on both sides yields $L(z_u) > L(z_v)$. $\qquad \square$

We next demonstrate that for the optimal predictor in the form of Lemma 4.2, the scaling factor $c$ is strictly within the interval $(0,1)$.

**Lemma 4.4.** *Let $z_c = c(Z_k + \xi)$, where $\xi \sim \mathcal{N}(0, v)$ with $v > 0$, and $\xi$ is independent of $Z_k$. The optimal scaling factor $c$ minimizing $L(z_c)$ satisfies $c \in (0,1)$.*

*Proof.* Define $S = Z_k + \xi$. Note that $S \sim \mathcal{N}(0, 1+v)$. We expand the loss $L(z_c)$:

$$L(z_c) = \mathbb{E} \left[ -\sigma(Z_k) \cdot cS + \log(1 + e^{cS}) \right].$$

Let $g(c) = L(z_c)$. Differentiating with respect to $c$:

$$g'(c) = \mathbb{E}[-S\sigma(Z_k) + S\sigma(cS)]$$
$$= \mathbb{E}[-Z_k\sigma(Z_k)] + \mathbb{E}[S\sigma(cS)].$$

The second derivative is $g''(c) = \mathbb{E}[S^2\sigma'(cS)]$. Since the sigmoid derivative is strictly positive, $g''(c) > 0$, implying that $g$ is strictly convex.

Evaluating the gradient at $c = 0$:

$$g'(0) = -\mathbb{E}[(Z_k + \xi)\sigma(Z_k)] + \mathbb{E}\left[S \cdot \frac{1}{2}\right]$$
$$= -\mathbb{E}[Z_k\sigma(Z_k)].$$

We observe that $\mathbb{E}[Z_k\sigma(Z_k)] = \text{Cov}(Z_k, \sigma(Z_k)) > 0$. Therefore, $g'(0) < 0$, which implies the minimizer of $g$ must lie to the right of 0.

Next, consider the gradient at $c = 1$:

$$g'(1) = -\mathbb{E}[Z_k\sigma(Z_k)] + \mathbb{E}[S\sigma(S)].$$

Define the function $h(u) = \mathbb{E}_X[X\sigma(X)]$ where $X \sim \mathcal{N}(0, u^2)$. We observe that $g'(1) = h(\sqrt{1+v}) - h(1)$. If $h(u)$ is increasing for $u > 0$, then $g'(1) > 0$, meaning the minimizer must be less than 1.

Using the reparameterization $X = uX'$ where $X' \sim \mathcal{N}(0,1)$, we write $h(u) = u \cdot \mathbb{E}_{X'}[X'\sigma(uX')]$. The derivative is:

$$h'(u) = \mathbb{E}_{X'}[X'\sigma(uX')] + \mathbb{E}_{X'}[u(X')^2\sigma'(uX')].$$

For $u > 0$, the first term is positive as it equals $\frac{1}{u}\text{Cov}(uX', \sigma(uX'))$. The second term is non-negative since the term inside the expectation is non-negative. Thus, $h'(u) > 0$ for $u > 0$. This concludes that the optimal $c$ lies in the interval $(0,1)$. $\qquad \square$

### 4.3. Connection to Excess Loss

Finally, we connect the variance of the logit estimator to the BCE loss to establish a lower bound on the excess loss.

**Theorem 4.5** (Lower Bound on Convergence). *Let $k$ denote the dimension of the feature space. Consider the feature distribution and network construction defined in Section 4.1. This construction satisfies the $M$-coverage condition with $M = k$. For the agent at the end of the pass $p$ (where $p \leq k - 1$), let $p^*$ denote the optimal global logistic predictor and $p_D$ (where $D = kp$) the predictor of the final agent. The excess loss is lower bounded by:*

$$L(p_D) - L(p^*) = \Omega\left(\frac{1}{p}\right) = \Omega\left(\frac{k}{D}\right).$$

*Proof.* We begin by relating the excess loss to the expected squared difference in the probability space. Invoking Lemma 3.3 and Lemma 3.4, the loss difference satisfies:

$$L(p_D) - L(p^*) = D(p^*||p_D) \geq 2\mathbb{E}\left[(p^* - p_D)^2\right].$$

The agent $A_D$ operates at the end of pass $p$. Let $z_D$ be the logit of this agent such that $p_D = \sigma(z_D)$. By Lemma 4.1, $A_D$ relies only on the information in $\mathcal{I}_p$. Consequently, by Lemma 4.2, $z_D$ takes the form:

$$z_D = c\left(Z_k + \frac{1}{\sqrt{p}}\xi\right), \tag{12}$$

for some constant $c$, where $\xi \sim \mathcal{N}(0, V_p)$. Since $z_D$ minimizes the BCE loss (Lemma 4.3), $V_p$ must be minimized. Lemma 4.2 states this minimum occurs at $V_p = 1$. Furthermore, Lemma 4.4 implies $c \in (0, 1)$.

To establish a lower bound, we relate the squared error in probabilities $(p^* - p_D)^2$ to the squared error in logits $(Z_k - z_D)^2$ using Mean Value Theorem. However, the sigmoid derivative vanishes for large inputs, which could dampen the probability difference even if the logit error is large. To address this, we restrict our analysis to a bounded region where the sigmoid function has non-vanishing derivative. Define the event $\mathcal{B}_R$ where both $Z_k$ and $\xi$ are bounded by $R$:

$$\mathcal{B}_R := \{|Z_k| < R, |\xi| < R\}.$$

By selecting a sufficiently large $R$, $\mathcal{B}_R$ captures a constant fraction of the probability mass. On this set, the arguments to the sigmoid function are bounded. By the Mean Value Theorem, there exists $\eta$ between $Z_k$ and $z_D$ such that

$$p^* - p_D = \sigma(Z_k) - \sigma(z_D) = \sigma'(\eta)(Z_k - z_D)$$

Since $|\eta| < 2R$ on $\mathcal{B}_R$, $\sigma'(\eta) \geq C_{\text{univ}} > 0$ for some constant $C_{\text{univ}}$ dependent only on $R$.

Restricting the expectation to $\mathcal{B}_R$ and defining $C_1 = 2C_{\text{univ}}^2$, we have:

$$2\mathbb{E}\left[(p^* - p_D)^2 \mathbf{1}_{\mathcal{B}_R}\right] \geq C_1 \mathbb{E}\left[(Z_k - z_D)^2 \mathbf{1}_{\mathcal{B}_R}\right].$$

Substituting the estimator form from Equation (12), we expand the quadratic term:

$$\mathbb{E}\left[\left((1 - c)Z_k - \frac{c}{\sqrt{p}}\xi\right)^2 \mathbf{1}_{\mathcal{B}_R}\right]$$

$$= (1 - c)^2 \mathbb{E}[Z_k^2 \mathbf{1}_{\mathcal{B}_R}] - \frac{2c(1 - c)}{\sqrt{p}}\mathbb{E}[Z_k \xi \mathbf{1}_{\mathcal{B}_R}]$$

$$+ \frac{c^2}{p}\mathbb{E}[\xi^2 \mathbf{1}_{\mathcal{B}_R}].$$

Since $Z_k$ and $\xi$ are independent and centered, and the region $\mathcal{B}_R$ is symmetric about the origin for both variables, $\mathbb{E}[Z_k \xi \mathbf{1}_{\mathcal{B}_R}] = 0$.

Since $Z_k$ and $\xi$ follow the same distribution on $\mathcal{B}_R$, we define $C_2 = \mathbb{E}[Z_k^2 \mathbf{1}_{\mathcal{B}_R}] = \mathbb{E}[\xi^2 \mathbf{1}_{\mathcal{B}_R}]$. This yields:

$$L(p_D) - L(p^*) \geq C_1 C_2 \left((1 - c)^2 + \frac{c^2}{p}\right).$$

The quadratic function $(1 - c)^2 + \frac{c^2}{p}$ is minimized at $c = \frac{p}{p+1}$. Substituting this value:

$$L(p_D) - L(p^*) \geq \frac{C_1 C_2}{p + 1} = \Omega\left(\frac{1}{p}\right).$$

Recalling that $p = D/k$, we conclude:

$$L(p_D) - L(p^*) = \Omega\left(\frac{k}{D}\right). \qquad \square$$

## Acknowledgements

This work is partially supported by DARPA expMath, ONR MURI 2024 award on Algorithms, Learning, and Game Theory, Army-Research Laboratory (ARL) grant W911NF2410052, NSF AF: Small grants 2218678, 2114269, and 2347322.

## Impact Statement

This paper presents work whose goal is to advance the field of Machine Learning. There are many potential societal consequences of our work, none of which we feel must be specifically highlighted here.

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

## A. Omitted Proofs

**Lemma 3.4.** *For the expected KL divergence $D(p\|q)$, the following inequality holds:*

$$D(p\|q) \geq 2\mathbb{E}\left[(p(x) - q(x))^2\right].$$

*Proof.* We verify the inequality pointwise for any $x \in \mathcal{X}$. We aim to show:

$$p(x)\log\frac{p(x)}{q(x)} + (1 - p(x))\log\frac{1 - p(x)}{1 - q(x)} \geq 2(p(x) - q(x))^2. \tag{13}$$

Define the function $f(p) = p\log\frac{p}{q} + (1 - p)\log\frac{1-p}{1-q} - 2(p - q)^2$. The first derivative with respect to $p$ is: $f'(p) = \log\frac{p}{q} - \log\frac{1-p}{1-q} - 4(p - q)$. The second derivative is $f''(p) = \frac{1}{p} + \frac{1}{1-p} - 4 = \frac{1}{p(1-p)} - 4$. For $p \in [0, 1]$, the term $p(1 - p)$ has a maximum value of $0.25$. Consequently, $\frac{1}{p(1-p)} \geq 4$, which implies $f''(p) \geq 0$. Since $f$ is convex and satisfies $f'(q) = 0$, the point $p = q$ is a global minimum. Observing that $f(q) = 0$, we conclude that $f(p) \geq 0$ for all $p$. Taking the expectation of both sides in (13) yields the result:

$$D(p\|q) = \mathbb{E}\left[p(x)\log\frac{p(x)}{q(x)} + (1 - p(x))\log\frac{1 - p(x)}{1 - q(x)}\right] \geq 2\mathbb{E}[(p(x) - q(x))^2]. \qquad \square$$

