# OpenReview forum: "Networked Information Aggregation for Binary Classification"
_ICML.cc/2026/Conference — ICML 2026 regular_

### Official Review · Reviewer_APeC · 2026-03-01

**Soundness:** 3
**Presentation:** 4
**Significance:** 3
**Originality:** 3
**Overall Recommendation:** 5
**Confidence:** 4

**Summary:**

The paper investigates the networked information aggregation problem, extending it from regression to classification. It provides an upper bound on the excess loss of O(M/√D) for paths of depth D satisfying an M-coverage condition, and a lower bound on the excess loss of Ω(k/D), where k is the feature dimension, demonstrating that network depth is a fundamental bottleneck in this setting.

**Compliance With Llm Reviewing Policy:**

Affirmed.

**Final Justification:**

I thank the authors for the explanation and additional results. I am adjusting my score to 5: Accept.

**Key Questions For Authors:**

Key Questions For Authors
1.	I am curious if this analysis can be easily extended to multi-class classification.
2.	The theoretical claims would be significantly strengthened by experimental validation, even on synthetic data.
a.	Are the assumptions (bounded moments, bounded coefficients, M-coverage condition) realistic and easy to verify in practice? Can you provide examples of real-world scenarios where these hold or fail?
b.	How does it perform compare to some basic baselines?
c.	Does it matter which agent in the network serves as the final predictor?
Minor comments
3.	I would add a reference explaining for the decentralized fraud detection application: ‘Despite these challenges, classification remains the primary modality for distributed applications ranging from medical diagnosis (Vepakomma et al., 2018) to decentralized fraud detection.’
4.	Lines 131-132 second column: ‘minimize the expected BCE loss’. I might be wrong, but I think that the expectation is already included in the definition of L.
5.	In the first equation in Definition 3.2. there seems to be one additional D_{KL}.

**Limitations:**

yes

**Strengths And Weaknesses:**

Soundness.
The theoretical analysis is rigorous, the proofs appear correct, and the assumptions seem reasonable. However, the work does not contain any experiments limiting the evaluation of a practical applicability of the claims.

Presentation.
The work is very well written and structured, making it easy to follow. I would just suggest to shorter the abstract and add a conclusion section at the end. If space is needed for the conclusion, consider moving some of the proofs to the appendix.
The authors clearly articulate their contributions and how they differ from prior work. They explain that extending the analysis from Kearns et al. (2026) from squared loss to logistic loss is nontrivial.

Significance.
The work addresses the relevant problem of networked information aggregation. It fills a gap, extending the analysis in Kearns et al. (2026) from squared loss to logistic loss which is not trivial, as BCE does not admit the variance decomposition that underlies the regression analysis. The influence to future applications is limited by the lack of experiments.

Originality.
The work analysis is original. It shows that the networked information aggregation problem for binary classification needs different proofs and techniques respect to the regression one. It shows that network depth is a bottleneck, necessary and not just sufficient.

---

> ### Author Rebuttal · Authors · 2026-03-31
>
> We thank the reviewer for the careful reading of our paper and for the positive assessment of its soundness, clarity, and originality. We also appreciate the thoughtful questions and suggestions, especially regarding experiments and practical applicability. These comments are very helpful for improving the paper.
>
>
> Regarding your questions:
>
>
> “I am curious if this analysis can be easily extended to multi-class classification.”
>
>
> We had not considered the multiclass case in this submission, but we agree that it is a natural direction. Our current view is that a similar approach may be possible, but we do not want to claim that it is immediate without checking the details carefully. We thank the reviewer for raising this point, and we think it is an interesting direction to explore further.
>
>
> “The theoretical claims would be significantly strengthened by experimental validation, even on synthetic data.”
>
>
> We agree. Following this suggestion, we ran initial experiments on the public binary-classification dataset Breast Cancer Wisconsin (Diagnostic) (via scikit-learn). An anonymized page containing only the plots is available here: [ https://anonymous.4open.science/r/Experiments-plots-5B3E/Plots.pdf ]. The first plot compares empirical lower- and upper- bounds with the actual BCE of the protocol on a simple path network, similar to our lower-bound path network. The second plot studies the effect of $M$ on the same simple path network. The last two plots report results for the same protocol on a fixed 50-node tree, shown in both top-down and bottom-up views; this tree is similar to the one considered in Networked Information Aggregation via Machine Learning (Kearns, Roth, and Ryu, p. 29).
>
>
> “Are the assumptions (bounded moments, bounded coefficients, M-coverage condition) realistic and easy to verify in practice? Can you provide examples of real-world scenarios where these hold or fail?”
>
>
> We appreciate this question and agree that the assumptions deserve further discussion. We will add a short discussion in the revision clarifying when these assumptions are natural and when they may be restrictive.
> The $M$-coverage condition is easy to verify once the feature assignment and network order are fixed. It is most natural in settings where informative features are distributed across nearby agents, and less natural in highly siloed systems where important features may be separated by long gaps in the network.
>
> The bounded-moment assumption is standard and often reasonable after preprocessing steps such as normalization or clipping, though it may fail for raw heavy-tailed features.
>
> The bounded-coefficient assumption is partly technical, but it is also a natural way to formalize a non-saturated logistic regime. Since $\sigma(z)=\frac{1}{1+e^{-z}}$ already spans a substantial range of probabilities for moderate values of \(z\), one does not need extremely large logits to represent meaningful uncertainty. Very large values of \(|z|\) correspond to probabilities extremely close to \(0\) or \(1\), i.e., nearly saturated predictions. Thus, if we exclude such nearly saturated regimes, it is natural to focus on a setting where the optimal logits remain bounded. Under standard feature normalization, a bounded coefficient norm is a convenient and interpretable way to encode this condition.
>
>
> “How does it perform compare to some basic baselines?”
>
>
> In our analysis, we compare the protocol's performance against a full-information logistic predictor. This is our primary benchmark because it represents the best possible prediction achievable if one had centralized access to all features.
>
>
> “Does it matter which agent in the network serves as the final predictor?”
>
>
>  Yes. On a path, later agents are never worse in expected BCE, since each agent can always pass through its parent’s logit unchanged, so the loss is non-increasing along the path. This is why the last agent is the natural output node in our theorem. In a more general DAG, different sinks may lie at the end of paths with different effective depth and coverage, so the choice of final predictor can matter.
>
>
> “I would add a reference explaining for the decentralized fraud detection application...”
>
>
>  Thank you for this suggestion. We agree that this sentence should be better supported, and we will revise the wording accordingly.
>
>
> “Lines 131-132 second column: ‘minimize the expected BCE loss’. I might be wrong, but I think that the expectation is already included in the definition of L.”
>
>
>  The reviewer is correct. We thank the reviewer for catching this.
>
>
> “In the first equation in Definition 3.2. there seems to be one additional D_{KL}​.”
>
>
> Thank you for catching this typo.
>
>
> If our clarifications have addressed your primary concerns, we would be grateful if you would consider updating your score.

---

> > ### Author Rebuttal · Reviewer_APeC · 2026-04-01
> >
> > I thank the authors for the explanation and additional results. I am adjusting my score to 5: Accept.

---

> > > ### Author Response · Authors · 2026-04-03
> > >
> > > Thank you so much for your time and suggestions.
> > >
> > > We noticed that while your comment kindly mentions adjusting the score to a 5 (Accept), the official score field on the original review hasn't updated yet. When you have a brief moment, could you please edit the numerical score in the review?
> > >
> > > Thank you again for your valuable input and your support of our paper.

---

### Official Review · Reviewer_epfH · 2026-03-09

**Soundness:** 4
**Presentation:** 4
**Significance:** 3
**Originality:** 3
**Overall Recommendation:** 5
**Confidence:** 4

**Summary:**

The authors consider a DAG of agents who use logistic regression to predict a binary label, based on some observables locally available to them together with the logits determined by their direct predecessors in the DAG. The key results are as follows: 1) Theorem 3.8 establishes that for an agent in the DAG such that there exists a path of length D leading to it, satisfying a coverage condition, namely any set of M consecutive agents on this path collectively hold the complete set of observables, then necessarily the excess loss at that agent, compared with the optimal logistic regression based on the full set of all observables, scales like O(M/sqrt(D)). Theorem 4.5 provides a result in the converse direction, stating that there exists a DAG such that, for a total number of k observables, a vertex at depth D in the DAG has an excess loss larger than Omega(k/D).

**Compliance With Llm Reviewing Policy:**

Affirmed.

**Key Questions For Authors:**

As above, could you provide specific motivation for the DAG architecture?

It seems that in Theorem 4.5, the construction you use satisfies the M-coverage assumption with M=k, but I did not see this mentioned. Maybe worth stating.

In Lemma 3.1 you implicitly assume the existence of an optimal logistic regressor; your result would probably hold without making this a hypothesis, but it might help if you could address this point explicitly.

In Definition 3.2, there seems to be a typo in D(p||q) (an extra D_KL on the right).

In the last steps of the proof of Lemma 3.6, it seems you use Lemma 3.3; this may be worth stating explicitly.

In the statement of Lemma 4.2, when you say 'minimizing the variance V_p yields V_p=1' reads odd, because it is not obvious that you cannot take V_p=0. Maybe a restatement saying that V_p depends on c would help.

**Limitations:**

yes

**Strengths And Weaknesses:**

The paper is very cleanly written, and motivates adequately the problem it studies. The modelling adopted provides a nice stylized formulation of the problem of collective classification from partially available observations, and communications among agents. The derivations are elegant. The results are also neat, highlighting the role of depth of agents in the communication DAG on the performance of their resulting classifiers. Maybe some additional motivation for why studying communications organized according to a DAG would strengthen the paper. It is clearly a theory paper, hence would appeal only to a subset of the ICML audience, but in this reviewer's view this is fine.

---

> ### Author Rebuttal · Authors · 2026-03-31
>
> Thank you for the careful reading and for the positive assessment of the paper. We appreciate that you found the paper clearly written and technically solid, and that you viewed the results as well motivated. We also appreciate the specific suggestions.
>
> First, regarding the motivation for the DAG architecture: we agree that this point deserves to be explained more explicitly. In part, this question already arises for the framework of Kearns et al., and one of our goals was to extend that perspective to a broader class of communication structures. The key intuition is that the DAG architecture can model a wide range of networked learning protocols by ``unrolling'' communication over time. Concretely, suppose the network consists of agents $A = \{A_1, A_2, \ldots, A_n\}$ and in round $r$ a subset of agents $A_r$ performs training and communicates with agents participating in round $r+1$. Then we can construct a DAG by creating a copy of each active agent in each round and connecting these copies according to the communication pattern between consecutive rounds. In this way, the depth of the DAG can be interpreted as the number of rounds of communication required by the protocol. For example, the path-based hard instance in Section~4 can also be viewed as arising from a cyclic communication network when it is unfolded over time. We will add this intuition to the paper to better motivate the DAG formulation.
>
> Second, regarding Lemma~3.1 and the existence of an optimal logistic regressor: Thank you for mentioning this point, yes this is one of our assumptions. We will address this point explicitly.
>
>
> Thank you for your other great points about the structure and presentation of the paper. We will continuously improve the writing for the camera-ready version and apply your comments.

---

> > ### Author Rebuttal · Reviewer_epfH · 2026-04-03
> >
> > No remaining issues for me.

---

### Official Review · Reviewer_3hGG · 2026-03-13

**Soundness:** 2
**Presentation:** 3
**Significance:** 2
**Originality:** 2
**Overall Recommendation:** 4
**Confidence:** 4

**Summary:**

This paper studies networked binary classification in a DAG, where each agent observes only a subset of features and logits from predecessor agents. Each agent fits a logistic model using BCE loss, and passes the logit onward. The paper extends the networked information-aggregation framework of Kearns et al. 2026 from squared-loss linear regression to logistic/BCE classification. The main claimed results are an upper bound and a lower bound. The upper bound show that under an M-coverage condition on a depth-D path, the final agent achieves excess loss $O(M/\sqrt{D})$, and the lower bound give an $\Omega(k/D)$ excess loss on a hard instance. The results show that, similar to the regression setting, depth is again the key bottleneck for information aggregation in the classification setting.

**Compliance With Llm Reviewing Policy:**

Affirmed.

**Final Justification:**

The rebuttal addressed my main concern about the correctness of the lower bound and the novelty of techniques. Although I still think the novelty of this work is somewhat limited due to the nature of the problem to be an extension of the previous work in regression setting, it is a neat contribution to the community.

**Key Questions For Authors:**

1. See W2. I am mostly unsure about the correctness of lemma 4.1.

**Limitations:**

Yes.

**Strengths And Weaknesses:**

Strengths:
1. The paper studies a natural and interesting extension of the Kearns et al. 2026 framework from regression to binary classification.
2. Both the upper-bound and lower-bound results are clean and well motivated.
3. The paper is clearly written and generally easy to follow.

Weaknesses:

My main concerns are twofold.
1. The novelty of this paper seems limited, which is extending the results of Kearns et al. 2026 to binary classification setting. However, it is somewhat well-known in the multi-accuracy/calibration literature that in classification setting, the loss function for the analysis should be replaced by KL loss, which makes the result less surprising. The technical proofs also appear to follow Kearns et al. 2026 quite closely. Although the result is still interesting, the main message is largely the same as in Kearns et al. 2026.
2. In the proof of Lemma 4.1, "Subsequent agents in the pass observe features independent of this updated residual" seems not correct. For example at the start of pass 3 one only knows
$z^{(2)} \in \operatorname{span}$ {$x_k, x_{k-1} $},  say  $z^{(2)} = c_0 x_k + c_1 x_{k-1}$.
After the first useful update  $z' = a x_{k-2} + b z^{(2)}$,  the next feature $x_{k-1}$ is generically still correlated with the updated residual, since
$\operatorname{Cov}(x_{k-1}, \eta) = -a - b c_0 + 2 b c_1$ after writing  $z' = (b c_0) Z_k + \eta$.

Minor issues:
1. Typos in def 3.2.
2. Although the paper mentions "finite dataset" in the abstract, the theorems are proved in population. It would be helpful to make it clear.

---

> ### Author Rebuttal · Authors · 2026-03-31
>
> We sincerely thank the reviewer for their positive assessment of our work and for finding the extension natural, the bounds well-motivated, and the paper clearly written.
>
> Your rigorous check of our proofs is highly appreciated, and you raised excellent points. Below is our response to your specific concerns, which we will incorporate into the camera-ready version.
>
>
> Correction to Lemma 4.1 Proof
>
> You are absolutely correct that the phrase "Subsequent agents in the pass observe features independent of this updated residual" is flawed, and your counterexample perfectly illustrates why. Thank you for catching this.
>
> To clarify the intent of Lemma 4.1: the purpose of this lemma is not to claim that the prediction stops updating after the first useful feature. Rather, its true meaning is strictly to establish that the prediction at the end of pass $p$ is exclusively a function of the restricted information set $I_p$.
>
> To fix the proof, we will change the problematic sentence to: "Subsequent agents in the pass observe only features in $I_p$." To explain why the lemma and the broader proof hold despite the correlation: In pass $p+1$, the agents sequentially observe features $x_1, \dots, x_k$. The first "useful" new feature they encounter is $x_{k-p}$. Once this feature is observed, the information set expands to $I_{p+1} = I_p \cup \{x_{k-p}\}$. The remaining agents in that specific pass then observe the features $x_{k-p+1}, \dots, x_k$. This subsequent sequence of features is exactly the set $I_p$.
>
> You are completely right that the logit and the predictions will change and improve as these subsequent features are processed. We fully acknowledge this, and we do not claim the prediction remains static. The crucial fact is that, regardless of how much these subsequent agents update the prediction, it remains strictly a function of the features in $I_{p+1}$.
>
> Lemma 4.2 holds for any linear predictor based on $I_{p+1}$. In the rest of the proof, we bound the error for the best possible prediction based on $I_{p+1}$. Because the actual prediction at the end of the pass—despite all of its improvements—is still fundamentally constrained to the set $I_{p+1}$, its performance cannot exceed that of the optimal restricted predictor. Thus, proving that any prediction based on $I_{p+1}$ is bad inherently proves that the real prediction is bad as well. We will update the proof text to explicitly clarify this logic.
>
>
> Addressing the Novelty Concern
>
>
> We agree that the use of KL divergence is standard when analyzing classification in the multi-accuracy and calibration literature. However, executing this within a sequential, networked DAG framework introduces distinct, non-trivial geometric challenges.
>
> As we highlight in Section 1.2 of our paper, bridging the "regression-to-classification gap" is a recognized, non-trivial challenge across multiple subfields. In areas ranging from sketching/subspace embeddings to conformal prediction, techniques that are elegant for squared-loss regression consistently require genuinely different tools and progress measures (like KL/Bregman divergences) when adapted for classification. Our work tackles this exact systemic gap within the context of constrained, decentralized learning.
>
> The analysis in Kearns et al. '26 fundamentally relies on the geometry of the squared loss—specifically, exact orthogonality of residuals and the Pythagorean variance decomposition. Because Binary Cross-Entropy (BCE) does not admit a comparable bias-variance decomposition, their proof techniques do not transfer directly. Furthermore, linear aggregation in probability space is not equivalent to linear aggregation in feature space, which forces our protocol to pass logits rather than probabilities.
>
> To overcome the lack of an exact Euclidean residual decomposition, our upper bound requires a genuinely different set of tools: we decompose the loss via an expected KL divergence (Lemma 3.3), lower-bound the KL divergence using Pinsker's inequality to bound prediction error (Lemma 3.4), and integrate this with a pigeonhole stability argument over the path (Theorem 3.8). We hope this clarifies why the transition from regression to classification in this specific networked setting represents a distinct and worthy theoretical contribution.
>
> Minor Issues:
>
>
> Thank you for mentioning the typos, especially the phrase “finite dataset” in the abstract. We will correct the issues in Definition 3.2 and other minor writing issues in the revised version.
>
>
>
> If our clarifications have addressed your primary concerns, we would be grateful if you would consider updating your score.

---

> > ### Author Rebuttal · Reviewer_3hGG · 2026-04-04
> >
> > Thank you for the response. I will increase my score accordingly.

---

### Official Review · Reviewer_5Tff · 2026-03-18

**Soundness:** 3
**Presentation:** 2
**Significance:** 3
**Originality:** 2
**Overall Recommendation:** 4
**Confidence:** 2

**Summary:**

In this paper, the authors consider social learning where agents are nodes of a directed acrylic graph. The direction of the edge indicated the direction of information flow. The goal is to learn the predictor for a binary classification problem where labels are related to the features through a logistic regression model. Each agent observes a subset of the features. The paper provides a protocol and its theoretical analysis (Theorem 3.8) which gives a bound of excess loss. They also give a lower bound (Theorem 4.5). This paper is an extension of Kearns et al'26 paper to the classification setup.

**Compliance With Llm Reviewing Policy:**

Affirmed.

**Final Justification:**

I still stand by my previous review and will keep the same score.

**Key Questions For Authors:**

- Why is it a good idea to use the same distribution as Kearns et al'26 in the Lower bound?
- Section 2.3: "To preserve the information geometry ..... final probabilities". I do not understand this sentence.
-page 5, left column, line 244: "..., since p_{i-1} is in a stricter subset of p_i". I am not sure what it means.

**Limitations:**

yes

**Strengths And Weaknesses:**

Strengths
- Even though the paper is an extension of Kerns et. al. 26 paper and uses similar ideas and similar distribution for lower bound, it still requires non-trivial effort to translate the ideas to logistic regression with cross entropy loss.
- There are no typos. The proofs are written well.

Weaknesses
- The writing can be greatly improved. I would have preferred the main results (Theorem 3.8 and 4.5) stated first and discussed. Right now they are buried after the lemma. Without reading the theorem statements, the readers are left clueless with lemma after lemma.
- The upper and lower bounds are not discussed together. I would have preferred some remarks about the gap.
- Section 2.3: "Let z_j  be the logit ....." Please formalise it.
- Section 2.3: It would be easier to read if the protocol is formalised like in an algorithm format. Right now it is not very clear.
- Page 2, line 1, left column: " We show that if .... all features". The sentence seems incomplete.
- Page 3, left column, last line: "...parent's logits". What is a logit? it has not been defined at this point.
- p_i(x) seems to be replaced with p_i arbitrarily. Please stick to one notation.

---

> ### Author Rebuttal · Authors · 2026-03-31
>
> We sincerely thank the reviewer for their careful reading of our manuscript and for recognizing the effort required to extend these networked learning guarantees to logistic regression and cross-entropy loss.
>
> Thank you for your great points about the structure and presentation of the paper. We will continuously improve the writing for the camera-ready version and apply your comments.
>
>
> Regarding your questions:
>
>
> “Why is it a good idea to use the same distribution as Kearns et al'26 in the Lower bound?“
>
> It is a very good question; we will add a short intuition for this distribution. The intuition is that we need to create an example that minimizes the dependency of the features of the agents. As you can see, the features of any two non-sequential agents are fully independent, and most of them are independent of the label as well. However, in the best prediction using all features, because of the simple linear relation between the features and the label, we can predict it very well. We need all features, and the lack of some features leads to a considerable error.
>
>
> "Section 2.3: "To preserve the information geometry ..... final probabilities". I do not understand this sentence."
>
> This sentence intends to say that agents pass the logit, which is a linear combination of the features, instead of the probabilities resulting from the sigmoid function. This helps the predictions remain a linear function of the features at any step. We will change this sentence to make it clearer.
>
>
> “page 5, left column, line 244: "..., since p_{i-1} is in a stricter subset of p_i". I am not sure what it means.”
>
> In the lemma 3.3, where $p^*$ denotes the optimal predictor on set $S$ and $q$ represents an arbitrary predictor on $S$: we intended to establish that this relationship applies to $p_{i-1}$ and $p_i$. Specifically, $p_{i-1}$ can be viewed as a prediction restricted to the unit information set $\{z_{i-1}\}$, which is a subset of the broader set $S$ from which $p_i$ is derived (since agent $A_{i-1}$ passes $z_{i-1}$ to $A_i$). We will refine the paper to make this structural dependency more explicit.
>
>
> If our clarifications have addressed your primary concerns, we would be grateful if you would consider updating your score.

---

> > ### Author Rebuttal · Reviewer_5Tff · 2026-04-01
> >
> > I will keep the same score.

---

### Decision · Program_Chairs · 2026-04-30

**Decision:**

Accept (regular)

**Comment:**

The paper builds on recent work by Kearns, Roth, and Ryu (2026) on information aggregation over a DAG where each node represents an agent which has a partial view of the features and the goal is to aggregate information by traversing the DAG sharing predictions to the neighbors. This paper extends their setup from squared loss regression tasks to binary classification with logistic loss by sharing "logits" instead of "predictions". This allows them to transfer the machinery to the binary case using  standard properties of the logistic loss.

The reviewers were positive about the paper overall, and thought the extension was broadly interesting. There were concerns about technical novelty given that the paper super closely follows Kearns, Roth, and Ryu (2026), but the reviewers felt this still required some non-trivial work, and the idea of exchanging the right information was novel. Therefore, I lean towards accept.

I encourage the authors to make sure to improve the writing, correct the proof errors, and add more clear explanations of how their work stands out from prior work in the camera-ready, if the paper is accepted.